# Randomised controlled trial comparing hydroxyapatite coated uncemented hemiarthroplasty with cemented hemiarthroplasty for the treatment of displaced intracapsular hip fractures: a protocol for the WHITE 5 study

Miguel Antonio Fernandez [1,2] Juul Achten,[1] Robin Gillmore Lerner [1] Katy Mironov,[1] Nicholas Parsons,[3] Melina Dritsaki,[4] May E Png,[4] Alwin McGibbon,[5] Jenny Gould,[5] Xavier Griffin [1] Matthew L Costa[1]

For numbered affiliations see end of article.

**Correspondence to**
Professor Matthew L Costa;
matthew.costa@ndorms.ox.ac.uk

## ABSTRACT

**Introduction** Hip fracture is a serious injury in adults, especially those aged over 60 years. The most common type of hip fracture (displaced intracapsular) is treated for the majority of patients with a partial hip replacement (hemiarthroplasty). The hemiarthroplasty implant can be fixed to the bone with or without bone cement. Cement is the current recommended technique but recently some risks have been identified, which could potentially be avoided by using uncemented implants. Controversy, therefore, remains about which type of hemiarthroplasty offers patients the best outcomes.

This is the protocol for a multicentre randomised controlled trial comparing cemented hemiarthroplasty versus uncemented hemiarthroplasty for patients 60 years and over with a displaced intracapsular hip fracture.

**Methods and analysis** Multicentre (a minimum of seven UK hospitals), multisurgeon, parallel group, two-arm, superiority, randomised controlled trial. Patients aged 60 years and older with a displaced intracapsular hip fracture treated with hemiarthroplasty surgery are eligible. Participants will be randomly allocated on a 1:1 basis to either a cemented hemiarthroplasty or a modern hydroxyapatite coated uncemented hemiarthroplasty. Otherwise all care will be in accordance with the National Institute for Health and Care Excellence guidance. A minimum of 1128 patients will be recruited to obtain 90% power to detect a 0.075-point difference in the primary endpoint: health-related quality of life (EuroQol 5 dimensions 5 levels) at 4 months postinjury. The treatment effect will be estimated using a two-sided t-test adjusted for age, gender and cognitive impairment based on an intention-to-treat analysis. Secondary outcomes include mortality, complications including revision surgery and cause, mobility status, residential status, health-related quality of life at 1 and 12 months and health resource use. A within-trial economic analysis will be conducted.

**Ethics, dissemination and funding** Wales Research Ethics Committee 5 approved the feasibility phase on 2 December 2016 (16/WA/0351) and the definitive trial on 22 November 2017 (17/WA/0383). This study is sponsored by the University of Oxford and funded by the National Institute for Health Research, Research for Patient Benefit (PB-PG-0215–36043 and PB-PG-1216–20021). A manuscript for a peer-reviewed journal will be prepared and the results shared with patients via local mechanisms at participating centres.

**Trial registration number** ISRCTN18393176

## Strengths and limitations of this study

► Pragmatic multicentre randomised controlled trial.
► Powered to detect differences in health-related quality of life.
► Inclusion of participants with and without cognitive impairment.
► Outcomes include the UK core outcome set for hip fracture.
► The trial will not capture late complications, implant failure or revision surgery beyond 1-year postinjury.

## INTRODUCTION

Hip fracture is one of the biggest challenges facing patients and healthcare systems. Every year, in the UK, there are more than 65 000 hip fractures.[1] Displaced 'intracapsular' fractures represent approximately half of all hip fractures. In these fractures, the head of the femur is broken off at the neck. The blood supply to the femoral head is tenuous and, even if the fracture is fixed back in to its anatomical position, fracture healing is unreliable and 40% of patients will have a failure of fixation. For this reason, arthroplasty (replacement) of the femoral head is preferred in the UK. A minority of patients receive a 'total hip arthroplasty' (replacing

the 'socket' of the hip joint as well as the femoral head) if they are particularly active before the fracture, but, in the most cases, only the head of the femur is replaced; a hemiarthroplasty.

The National Institute for Health and Care Excellence (NICE)[2] guidelines for hip fracture recommend the use of bone cement to aid in the fixation of the hemiarthroplasty implant to the bone. These guidelines are based on outcomes of studies, which show less postoperative pain and better function when cemented components were used compared with their first generation uncemented counterparts (eg, the Austin Moore prosthesis). The evidence base for these guidelines is summarised in a Cochrane review,[3] which acknowledged the lack of evidence for more contemporary uncemented implants and concluded that 'there is still a place for trials of contemporary uncemented stems with cemented stem hemiarthroplasty'. A recent independent systematic review and meta-analysis compared cemented hemiarthroplasty with modern uncemented hemiarthroplasty and concluded: 'there remains a need for a methodologically sound, large muticentre RCT comparing modern cemented and cementless hemiarthroplasty stems in the medium—and long-term, not only focusing on mortality and complications but also on patient reported outcome measures.'[4]

This is the protocol for a multicentre parallel-group randomised controlled trial that compares cemented hemiarthroplasty and modern hydroxyapatite coated uncemented hemiarthroplasty.

## AIMS AND OBJECTIVES

The aim of this randomised controlled trial is to compare health-related quality of life (HRQoL) in participants over 60 years of age with a displaced intracapsular hip fracture receiving a modern hydroxyapatite coated uncemented hemiarthroplasty versus the current standard-of-care cemented hemiarthroplasty.

### The primary objective is

► To quantify and draw inferences on observed differences in participants' HRQoL between the trial treatment groups at 4 months postinjury.

### The secondary objectives are

► To quantify and draw inferences on the observed differences in participants' HRQoL between the trial treatment groups at 1 and 12 months postinjury.
► To quantify and draw inferences on the observed differences in the proportion of complications, including further surgery, within the first 12 months postinjury between the trial treatment groups.
► To quantify and draw inferences on observed differences in mortality, mobility and residential status at 1, 4 and 12 months postinjury.
► To investigate, using appropriate statistical and economic analytical methods, the resource use, costs

and incremental cost effectiveness of the trial treatment groups at 4 months postinjury.

## METHODS AND ANALYSIS
### Study design

A multicentre, parallel group, two-arm, superiority, standard-of-care randomised controlled superiority trial in a minimum of seven UK recruitment centres. Each of these recruitment centres is recruiting hip fracture patients as part of the World Hip Trauma Evaluation hip fracture cohort study.[5] The study is conducted in two phases: an initial feasibility phase in which the acceptability of the interventions and trial processes were tested, and a definitive phase which comprises the main trial. Data from the feasibility phase will be locked, and not analysed, at completion. At the end of the definitive main trial phase, data from the two phases will be analysed together as a single dataset.

### Patient and public involvement

At the centre of this work is the potential for patient benefit by reducing the risks of hip fracture surgery and improving patient outcomes. The involvement of patient and public representatives has, therefore, been a key element of this work to ensure that the running of the trial remains patient focused. JG and AM are patient representative coauthors who have been involved in all aspects of this work from designing the trial, obtaining grant funding, developing the trial materials and participation in the running of the trial. More specifically, the role of patient representatives has been to ensure that trial processes and information are focused on patients, to provide the patient perspective as members of the trial management group, and to ensure that all trial processes have taken into account the needs of patients both with and without cognitive impairment.

### Eligibility

Patients will be screened against the following criteria:

#### Inclusion criteria

► All patients, both those with and without capacity, presenting with a displaced intracapsular fracture of the hip suitable for hemiarthroplasty.

#### Exclusion criteria

► Patients younger than 60 years of age.
► Patients who are managed non-operatively.
► Patients who are treated with a total hip replacement.

### Consent

Patients with a hip fracture are a clinical priority for urgent operative care. They undergo surgery on the next available trauma operating list. All patients with a fracture of the hip are in pain and will have received opiate analgesia. It is, therefore, understandable that the majority of patients find the initial period of their treatment in hospital confusing and disorientating. Similarly, patients' next of kin, carers and friends are often anxious

at this time and may have difficulty in absorbing the large amounts of information that they are given about the injury and plan for treatment. In this emergency situation, the focus is on obtaining consent for surgery (where possible) and on informing the patient and any next of kin about immediate clinical care. It is often not possible for the patient, relative or carer (consultee) to review trial documentation, consider the information and communicate an informed decision about whether they would wish to participate in the study. The consent procedure for this trial will reflect that of the surgery, with the clinical team assessing capacity before taking consent for the surgical procedure, and this capacity assessment then being used to guide the proper approach to consenting to the research. An appropriate method, in line with the mental capacity act and approved by the National Research Ethics Committee, will be used to gain either prospective or retrospective consent from the patient or appropriate consultee by a Good Clinical Practice (GCP) trained, appropriately delegated member of the research team.

## Randomisation and blinding

The allocation sequence will be computer generated by the trial statistician and administered, using secure online randomisation, by Oxford Clinical Trials Research Unit (OCTRU), University of Oxford. Participants will be enrolled by the operating surgeon or trial research associate. The research associate will receive notification of the allocated treatment and inform the surgeon and the operating theatre staff. The randomisation algorithm will use variable block sizes (2, 4 and 6), to ensure good treatment balance, at each of the recruitment centres. The surgery will be performed under the care of the consultant surgeon on-call at the time the patient is admitted as per local practice in the recruitment centres. The large number of surgeons—previous experience in similar trials suggests over 200 surgeons will take part—and the wide skill mix should minimise any surgeon-specific effects such that stratification by surgeon is unnecessary. In order to negate bias in the self-reported HRQoL outcome measures, participants will be blinded to treatment allocation. The operating surgeon cannot be blinded to the allocation but they will not be involved in the assessment of outcomes. Patients will be blinded until the completion of the trial when the blinding will be broken if requested by the patients. There will be no formal analysis of the success of the blinding.

## Postrandomisation withdrawals and exclusion

Throughout the study, screening logs will be kept to determine the number of patients assessed for eligibility and reasons for any exclusion. Participants may withdraw from the study at any time without prejudice. If a participant withdraws from the study completely, data collected up until the point of withdrawal will be included in the final analysis. Patients who decline to continue to take part will be given the opportunity to discuss and inform the research team of the reasoning behind their decision not to take part. Similarly, data obtained from participants who die before they or a personal consultee has been approached about continued consent/agreement will be included in the final analysis.

## Treatments
### Standardised treatment pathway

Participants will usually be assessed in the emergency department. Diagnosis of a hip fracture will be confirmed by a plain radiograph, as per routine clinical care. Supplementary imaging will be at the discretion of the treating clinical team. Routine investigations, anaesthetic assessment, antibiotic and venous thromboembolic prophylaxis will be used as per local policy. A regional or general anaesthesia technique will be used for every participant as per routine clinical care. All participants will receive perioperative prophylactic antibiotics in accordance with current protocols agreed at each recruitment centre. Postoperative analgesia will be prescribed intraoperatively and reviewed by the responsible clinical teams as appropriate.

In the postoperative period, as per standard of care, all participants will undergo a physiotherapy and occupational therapy assessment to create a rehabilitation plan. The aim of this plan will be for participants to mobilise through early, active and full weight bearing. Participants will be discharged from the acute Orthopaedic Trauma Ward at the earliest safe opportunity to the most appropriate discharge destination as determined by the multidisciplinary clinical team. Participants will be given the relevant National Health Service (NHS) Trust information packs.

### Interventions

Appropriate preparation, positioning and surgical technique will be left to the discretion of the operating surgeon, as per their normal clinical practice. Participants will be randomly allocated to one of two treatment groups:

Group 1: Cemented hemiarthroplasty. Replacement of the femoral head and neck with a cemented femoral stem and head as per NICE guidance; the current standard-of-care (control) intervention.

Group 2: Modern uncemented hemiarthroplasty. Replacement of the femoral head and neck with a modern (hydroxyapatite coated) uncemented hemiarthroplasty; the alternative intervention.

Both types of hip replacement are used routinely throughout the NHS. While the principles of both cemented and uncemented arthroplasty are inherent in the technique, the surgical approach and other technical aspects of the surgery will be left to the discretion of the senior operating surgeon as per their normal practice. This will ensure that the results of this pragmatic trial can be extrapolated broadly across the NHS.

### Outcomes and data management

Personal data collected during the study will be handled and stored in accordance with the 1998 Data Protection

Act, which requires data to be anonymised as soon as it is practical to do so. The data collected from participants will be entered in linked-anonymised form to the trial database. All electronic patient-identifiable information will be stored on a secure, password-protected database at the University of Oxford, accessible only to the research team. Any paper copies of identifiable data, and corresponding reidentifying links to the participant trial ID, will be stored separately, in a locked cabinet in an access-restricted part of the Kadoorie Centre, John Radcliffe Hospital, Oxford. The study databases will be developed by the trial programmer and all specifications agreed between the trial programmer, statistician and trial manager and other relevant members of the trial team. The procedure for data entry will be documented in the data management plan. As per routine clinical care, the existing National Hip Fracture Database (NHFD) dataset will be collected via telephone interview or postal questionnaire. We will send data collected at follow-up to the NHFD via a secure email account for them to upload using the participant's date of birth and NHS number as identifiers.

### Primary outcome measure

The study primary outcome measure is EuroQol 5 dimensions 5 levels (EQ-5D-5L) index score at 4 months postinjury. The index score is derived from the five health state domains using the crosswalk value data sets.[6] EQ-5D-5L is a validated, self-rated instrument comprising a Visual Analogue Scale (VAS) measuring health and a health status instrument, consisting of a five-level response (no problems, some problems, moderate problems, severe problems and unable) for five domains related to daily activities[7]: (1) mobility, (2) self-care, (3) usual activities, (4) pain and discomfort and (5) anxiety and depression. The EQ-5D-5L instrument[7] facilitates the generation of a utility score from a person's HRQoL, which refers to the preference that individuals have for any particular set of health outcomes. As per the current NICE position statement, the responses to the EQ-5D-5L will be converted into multiattribute utility scores using an approved 'cross-walk' to the 3L instrument and its established time trade-off utility algorithm for the UK population.[8] A respondent's EQ-VAS gives self-rated health on a scale where the endpoints are labelled 'best imaginable health state' (100) and 'worst imaginable health state' (0). EQ-5D is part of the UK Core outcome set[9] and has been shown to be responsive to change,[10 11] including when reported by proxy for those with cognitive impairment.[12 13]

### Secondary outcome measures

HRQoL (EQ-5D-5L) at 1 and 12 months postinjury. Mortality, revision surgery and cause, and all complications will be obtained from the patients' medical record. Patient interviews and questionnaires will be used to assess mobility status (walking ability indoors and outdoors) and residential status. An assessment of resource use will inform the economic analysis plan below.

**Table 1** Group sizes required following sample size calculations

| Power, % | MCID | | |
| --- | --- | --- | --- |
| | 0.07 | 0.075 | 0.08 |
| 80 | 290 | 253 | 222 |
| 90 | 387 | 338 | 297 |

MCID, minimal clinically important difference.

### Sample size

The best available evidence we have from data collected during previously performed studies in this patient population suggests that the SD for EQ-5D-5L at 4 months postinjury is approximately 0.3 points.[10] The best available evidence for what constitutes a minimal important difference (MID) for EQ-5D comes from a review of MID estimates.[14] A review of the literature shows an estimated median value of 0.08 for the MID for EQ-5D. Using our previously established standard deviation (SD=0.33), this suggests a standardised effect size of approximately 0.24; a 'small to moderate effect' based on Cohen's criteria.[15] Assuming that the EQ-5D at 4 months postinjury has an approximate normal distribution, which Parsons et al[10] suggest is reasonable, and a 1:1 allocation ratio, then if the true difference between the experimental and control group EQ-5D means is in the range 0.07–0.08, we will need to recruit the below number of participants (see table 1) in each group to be able to reject the null hypothesis that the population means are equal with probability (power) 0.8 and 0.9 and type I error rate of 5% (significance).

Taking the intermediate minimal clinically important difference of 0.075, for 80% (90%) power, we would need to recruit 253 (338) patients in both the experimental arm and in the control arm, 506 (676) in total. In this population we expect considerable lost to follow-up due mainly to patients declining consent to further follow-up and incapacity, so we have assumed that only 60% of recruited study participants will be available at the definitive endpoint at 4 months. This gives a total sample size of 844 for 80% power and 1128 for 90% power. Conservatively, we aim to recruit 1128 to ensure 90% power based on these assumptions.

### Statistical analysis

The main analysis will investigate differences in the primary outcome measure, EQ-5D (HRQoL), between the two treatment groups on an intention-to-treat (ITT) basis at 4 months postinjury on a complete-case basis. The 'death-adjusted' ED-5D measure, imputing death as zero, is the preferred analysis option in this population, and will be used for all analyses.[16] An initial analysis will test for differences between treatment groups using linear regression analysis, based on a Normal approximation for EQ-5D. Tests will be two sided and considered to provide evidence for a significant difference if p values are less than 0.05 (5% significance level). Estimates of treatment effects will be presented with 95% CIs.

The stratification used for randomisation will ensure balance in treatment allocation across recruitment centres. In addition to the unadjusted analyses, we will also undertake regression analyses to adjust for any imbalance between treatment groups in patient baseline (preinjury) EQ-5D, age and gender. Age and gender are likely to be reported by all study participants, however, if baseline EQ-5D is poorly reported, such that it reduces the available sample size for the definitive analysis, then it will not be included in the definitive model. The fixed-effects analysis (linear regression model) will be generalised by adding a random effect for recruitment centre to allow for possible heterogeneity in patient outcomes due more generally to the recruitment centre. The adjusted mixed-effects regression will be the definitive (primary) analyses and will be reported as such. Analyses will be undertaken using the specialist mixed-effects modelling functions available in the software package R (http://www.r-project.org/).[17] EQ-5D data will be assumed to be approximately normally distributed; possibly after appropriate variance-stabilising transformation. The primary focus will be the comparison of the two treatment groups of patients, and this will be reflected in the analysis which will be reported together with appropriate diagnostic plots that check the underlying model assumptions. Results will be presented as mean differences between the trial groups, with 95% CIs. The primary analysis will be on an ITT basis. However, a per-protocol analysis will also be undertaken by repeating the primary outcome analysis using the actual treatment received by participants (in contrast to the allocated treatment) as a sensitivity analysis to examine the robustness of conclusions to different assumptions about departures from the stated protocol.

In addition to the cross-sectional analysis at 4 months, we will also perform a longitudinal analysis of EQ-5D data collected at all study time points (1, 4 and 12 months) to a single value, namely the area under the curve (AUC) and facilitated comparisons of the AUCs between treatment groups. It is advisable in the presence of missing data to use summary statistics generated by mixed models, as estimates will not be biased under the assumption that data are missing at random, or completely at random. Therefore, we will fit a repeated measures mixed model, with the same fixed effect structure as used in the primary analyses, but with a three level random effects structure where observations (time points) are nested within participants, and participants nested within recruitment centre.

It seems likely that some data may not be available due to voluntary withdrawal of patients, lack of completion of individual data items or general lost to follow-up. Where possible the reasons for data 'missingness' will be ascertained and reported. Although missing data are not expected to be a problem for this study, the nature and pattern of the missingness will be carefully considered—including in particular whether data can be treated as missing at random (MAR). If judged appropriate, missing data will be imputed using the multiple imputation facilities available in R.[18] Any imputation methods used for scores and other derived variables will be carefully considered and justified. If the degree of missingness is relatively low, as expected, the primary analysis will be based on complete cases only (complete-case analysis), with analysis of imputed datasets used to assess the sensitivity of the analysis to the missing data. Reasons for ineligibility, non-compliance, withdrawal or other protocol violations will be stated and any patterns summarised. More formal analysis, for example, using logistic regression with 'protocol violation' as a response, may also be appropriate and aid interpretation.

Secondary analyses will be undertaken using the strategy described for the primary outcome for other approximately normally distributed outcome measures; for example, EQ-5D at 12 months. For dichotomous outcome variables, such as indicators of adverse events and other complications related to the trial interventions, mixed-effects logistic regression analysis will be undertaken with results presented as odds ratios (and 95% CIs) between the trial groups. The temporal patterns of any complications will be presented graphically and if appropriate a time-to-event analysis (Kaplan-Meier survival analysis) will be used to assess the overall risk and risk within individual classes of complications (eg, revision) and death. Cox's proportional hazards regression will be used to test for differences in death rates between the trial intervention groups, after adjusting for age and gender. Multiple complications will be defined as two or more independent events, that is, not continuations of a previous complication, for the same patient and will be identified only after discussion with the clinical team.

## Economic analysis

A within-trial cost-effectiveness analysis of cemented versus uncemented hemiarthroplasty in participants over 60 years of age with a displaced intracapsular hip fracture will be conducted from the UK NHS and Personal Social Services perspective[19] in the base case analysis. Resource utilisation involving differences in surgical treatments between the two intervention groups will be obtained from case report forms (CRFs) that would be completed by the local research teams. Broader resource utilisation will be captured through CRFs and patient questionnaires administered at baseline, 1 month, 4 months and 12 months postinjury. Unit costs for health and social care resources will largely be derived from the latest available local and national sources and estimated in line with best practice. Costs will be standardised to current prices where appropriate. An incremental cost-effectiveness analysis, expressed in terms of incremental cost per quality-adjusted life year gained, will be performed. Results will be presented using incremental cost-effectiveness ratios (ICERs), net monetary benefit and cost-effectiveness acceptability curves generated via non-parametric bootstrapping. Multiple imputation

methods will be used to impute missing data and avoid biases associated with complete-case analysis. Sensitivity analyses involving economic analysis from the societal perspective and extending the time frame from 4 months to 1 year will also be conducted.

## Trial organisation and oversight

The day-to-day management of the trial will be the responsibility of the trial manager, based at the Nuffield Department of Orthopaedics, Rheumatology and Musculoskeletal Sciences and supported by OCTRU staff. This will be overseen by a trial management group, who will meet monthly to assess progress. It will be the responsibility of the trial manager to undertake training of the research associates at each of the study centres. The study statistician and health economist will be closely involved in setting up data capture systems, design of databases and clinical reporting forms.

A trial steering committee and an independent DAMOCLES[20] compliant data and safety monitoring committee, that will assess progress, conduct and participant safety, will be set up at the start of the study.

## Quality control

Quality control procedures will be undertaken during the recruitment and data collection phases of the study to ensure research is conducted, generated, recorded and reported in compliance with the protocol, GCP and ethics committee. The chief investigator and the trial manager will develop data management and monitoring plans.

## Ethics and dissemination

The results of this trial will be disseminated to the hip fracture clinical community via presentations at national and international meetings as well as publication in peer-reviewed journals. Authorship will be determined in accordance with the International Committee of Medical Journal Editors guidelines, and other contributors will be acknowledged. All data will be presented such that no individual participants can be identified. A lay summary informing patients and the public of the trial outcome will be available on the trial website. Further documentation suitable for the general patient and public communities will be prepared where appropriate by the research team in collaboration with lay representatives.

**Author affiliations**
[1]Oxford Trauma, NDORMS, University of Oxford, Oxford, UK
[2]Trauma & Orthopaedic Surgery, University Hospitals Coventry and Warwickshire NHS Trust, Coventry, UK
[3]Statistics and Epidemiology Unit, Warwick Medical School, University of Warwick, Coventry, UK
[4]Oxford Clinical Trial Unit, NDORMS, University of Oxford, Oxford, UK
[5]Patient Representative, London, UK

**Contributors** MLC, JA, NP, MAF, MD, AM, JG and XG were responsible for obtaining grant funding for this trial. MAF, MLC, JA, RGL, KM, MEP, AM, JG, NP and XG developed the trial protocol and contributed to writing the manuscript. NP developed the statistical analysis plan and is leading the statistical analysis for the study. MD and MEP developed the economic analysis plan and MEP is leading the economic analysis for the study. All authors reviewed and agreed the final manuscript.

**Funding** This project is funded by the National Institute for Health Research (NIHR), Research for Patient Benefit (PB-PG-0215-36043 and PB-PG-1216-20021) and supported by the NIHR Oxford Biomedical Research Centre.

**Disclaimer** The views expressed in this article are those of the author(s) and not necessarily those of the NIHR, or the Department of Health and Social Care.

**Competing interests** MLC is a National Institute for Health Research (NIHR) Senior Investigator. He has ongoing expert consultancy with several companies; none involve the development of any implant for use in hip fracture care. XG is funded by a National Institute for Health Research Clinician Scientist Grant. Further funding from industry and charitable grants are and have been made available to his institution. All decisions relating to the design, conduct, analysis, write-up and publication of research are independent of these funders. He has ongoing expert consultancy with several companies; none involve the development of any implant for use in hip fracture care.

**Patient consent for publication** Not required.

**Ethics approval** Wales Research Ethics Committee 5 approved the feasibility phase on 2 December 2016 (16/WA/0351) and the definitive trial on 22 November 2017 (17/WA/0383).

**Provenance and peer review** Not commissioned; externally peer reviewed.

**ORCID iDs**
Miguel Antonio Fernandez http://orcid.org/0000-0003-1533-8752
Robin Gillmore Lerner http://orcid.org/0000-0002-8372-8181
Xavier Griffin http://orcid.org/0000-0003-2976-7523

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
