## [Reviewer comments · BMJ Open]

ARTICLE DETAILS

TITLE (PROVISIONAL)	Randomised controlled trial comparing hydroxyapatite coated uncemented hemiarthroplasty with cemented hemiarthroplasty for the treatment of displaced intracapsular hip fractures: a protocol for the WHITE 5 study
AUTHORS	Fernandez, Miguel; Achten, Juul; Lerner, Robin; Mironov, Katy; Parsons, Nicholas; Dritsaki, Melina; Png, May ee; McGibbon, Alwin; Gould, Jenny; Griffin, Xavier; Costa, Matthew

VERSION 1 – REVIEW

REVIEWER	Sebastian Mukka Department of Surgical and perioperative sciences Umeå University Sweden
REVIEW RETURNED	14-Oct-2019

GENERAL COMMENTS	Thank you for the opportunity to review the present study protocol. The study aims at including above 1000 patients for comparing uncemented and cemented hemiarthroplasty for displaced femoral neck fractures. The rationale of the study are important and valuable for the orthopedic community. The study is include a methodology at a scientific high level. However I have a number of queries: 1. The authors do not include any disease specific hip score system, please motivate the decision?2. The authors have chosen not to perform a test of blinding success, how come?3. Inclusion and exclusion criteria: Are pathological fractures included? Or please add as an exclusion criteria. Are patients with bilateral fractures included or only the first fracture? That would affect the choice of statistical methods used.4. In the introduction the authors could present the potential benefits for the intervention (cements implants) regarding short term mortality (<1week) and surgical time and the potential setback with early postoperative periprosthetic femur fractures. In general a well written and methodologically sound study.
--

VERSION 1 – AUTHOR RESPONSE

1. The authors do not include any disease specific hip score system, please motivate the decision?

Our research group has invested considerable efforts exploring the most appropriate outcome tool for patients with a hip fracture which not only captures the domains of health considered important by the patients themselves,¹ but which also enables inclusion of patients with cognitive impairment which is present in up to 40% of the hip fracture population.² The EQ-5D has emerged as the most appropriate tool to measure outcomes in hip fracture patients.³ It has been included in the core outcome sets of all three of the published consensus studies for hip fracture.⁴⁻⁶ Moreover, EQ-5D has been shown to be responsive,^{7,8} including when reported by proxy for those with cognitive impairment,⁹ and is as sensitive to change as the Oxford Hip Score (OHS) in this group of patients.⁷ Hip scores, such as the OHS, are not validated for completion by proxy and therefore could not be used in the important sub-group of hip fracture patients with cognitive impairment. In addition, the EQ-5D will be used for health economic analysis to calculate quality adjusted life years (QALYs), which is an important aspect of this definitive multi-centre randomised controlled trial.

1. Griffiths F, Mason V, Boardman F, Dennick K, Haywood K, Achten J, et al. Evaluating recovery following hip fracture: a qualitative interview study of what is important to patients. *BMJ Open* 2015;5(1):e005406.
2. Seitz DP, Adunuri N, Gill SS, Rochon PA. Prevalence of Dementia and Cognitive Impairment Among Older Adults With Hip Fractures. *J Am Med Dir Assoc* 2011;12(8):556–564.
3. Fernandez MA, Griffin XL, Costa ML. Hip fracture surgery: improving the quality of the evidence base. *The Bone and Joint Journal* 2015;97-B(7):875– 879.
4. Liem IS, Kammerlander C, Suhm N, Blauth M, Roth T, Gosch M, et al. Identifying a standard set of outcome parameters for the evaluation of orthogeriatric co-management for hip fractures. *Injury* 2013;44(11):1403– 1412.
5. Haywood KL, Griffin XL, Achten J, Costa ML. Developing a core outcome set for hip fracture trials. *The Bone and Joint Journal* 2014;96-B(8):1016–1023.
6. Hutchings L, Fox R, Chesser T. Proximal femoral fractures in the elderly: How are we measuring outcome? *Injury* 2011;42(11):1205–1213.
7. Parsons N, Griffin XL, Achten J, Costa ML. Outcome assessment after hip fracture: is EQ-5D the answer? *Bone and Joint Research* 2014;(3):69–75. 8. Tidermark JJ, Bergström GG. Responsiveness of the EuroQol (EQ-5D) and the Nottingham Health Profile (NHP) in elderly patients with femoral neck fractures. *Qual Life Res* 2007;16(2):321–330.

2. The authors have chosen not to perform a test of blinding success, how come?

This type of analysis is of limited benefit in this trial. It is not possible for surgeons to be blinded to the intervention but they will not be involved in the collection of outcomes. Patients and the research team collecting outcomes data are blinded to the intervention. The exact details of the implant used are only documented in the operation note and therefore unlikely to compromise the success of blinding.

3. Inclusion and exclusion criteria: Are pathological fractures included? Or please add as an exclusion criteria. Are patients with bilateral fractures included or only the first fracture? That would affect the choice of statistical methods used.

Pathological fractures are not formally excluded but the vast majority are treated with implants other than a hemiarthroplasty (e.g. intramedullary device, total hip replacement, proximal femoral

replacement) and would therefore not meet the inclusion criteria. Patients with bilateral displaced intracapsular hip fractures are not excluded.

4. In the introduction the authors could present the potential benefits for the intervention (cements implants) regarding short term mortality (<1week) and surgical time and the potential setback with early postoperative periprosthetic femur fractures.

We plan to include are more comprehensive introduction with the formal reporting of the trial results and therefore have not expanded the introduction for the protocol.

VERSION 2 – REVIEW

REVIEWER	Sebastian Mukka Department of surgical and perioperative sciences, Umeå University, Sweden
REVIEW RETURNED	12-Nov-2019
GENERAL COMMENTS	The authors have motivate their choices in research methodology. The study is well planned, different researchers might have opinions regarding each of the authors choice. The study protocol could be published in my opinion.